# ToM2C: Target-oriented Multi-agent Communication and Cooperation with Theory of Mind

**Yuanfei Wang**[\* 1, 4]**, Fangwei Zhong**[\* 2, 5]**, Jing Xu**[1]**, Yizhou Wang**[3]

[1] Center for Data Science, Peking University
[2] School of Artificial Intelligence, Peking University
[3] Center on Frontiers of Computing Studies, School of Computer Science, Peking University
[4] Adv. Inst. of Info. Tech, Peking University
[5] Beijing Institute for General Artificial Intelligence (BIGAI)
{ yuanfei_wang, zfw, jing.xu, yizhou.wang }@pku.edu.cn

## Abstract

Being able to predict the mental states of others is a key factor to effective social interaction. It is also crucial for distributed multi-agent systems, where agents are required to communicate and cooperate. In this paper, we introduce such an important social-cognitive skill, *i.e.* Theory of Mind (ToM), to build socially intelligent agents who are able to communicate and cooperate effectively to accomplish challenging tasks. With ToM, each agent is capable of inferring the mental states and intentions of others according to its (local) observation. Based on the inferred states, the agents decide "when" and with "whom" to share their intentions. With the information observed, inferred, and received, the agents decide their sub-goals and reach a consensus among the team. In the end, the low-level executors independently take primitive actions to accomplish the sub-goals. We demonstrate the idea in two typical target-oriented multi-agent tasks: cooperative navigation and multi-sensor target coverage. The experiments show that the proposed model not only outperforms the state-of-the-art methods on reward and communication efficiency, but also shows good generalization across different scales of the environment.

## 1 Introduction

Cooperation is a key component of human society, which enables people to divide labor and achieve common goals that could not be accomplished independently. In particular, humans are able to form an ad-hoc team with partners and communicate cooperatively with one another (Tomasello, 2014). Cognitive studies (Sher et al., 2014; Sanfey et al., 2015; Etel & Slaughter, 2019) show that the ability to model others' mental states (intentions, beliefs, and desires), called Theory of Mind (ToM) (Premack & Woodruff, 1978), is important for such social interaction. Consider a simple real-world scenario (Fig. 1), where three people (Alice, Bob and Carol) are required to take the fruits (apple, orange and pear) following the shortest path. To achieve it, the individual should take four steps sequentially: 1) observe their surrounding; 2) infer the observation and intention of others; 3) communicate with others to share the local observation or intention if necessary; 4) make a decision and take action to get the chosen fruits without conflict. In this process, the ToM is naturally adopted in inferring others (Step 2) and also guides the communication (Step 3).

In this paper, we focus on the **Target-oriented Multi-Agent Cooperation (ToMAC)** problem, where agents need to cooperatively adjust the relations among the agents and targets to reach the expectation, *e.g.,* covering all the targets (Xu et al., 2020). Such problem setting widely exists in real-world applications, *e.g.,* collecting multiple objects (Fig. 1), navigating to multiple landmarks (Lowe et al., 2017), following pedestrian (Zhong et al., 2021b), and transporting objects (Tuci et al., 2018). While running, the distributed agents are required to concurrently choose a subset of interesting targets and optimize the relation to them to contribute to the team goal. In this case, the key to realizing high-quality cooperation is to *reach a consensus among agents* to avoid the inner-team conflict. However, it is still difficult for the off-the-shelf multi-agent reinforcement learning methods, as they only implicitly model others in the hidden state and are inefficient in communication.

Here we propose a **Target-oriented Multi-agent Communication and Cooperation mechanism (ToM2C)** using Theory of Mind. Shown as Fig. 2, each agent is of a two-level hierarchy. The

---

\* indicates equal contribution

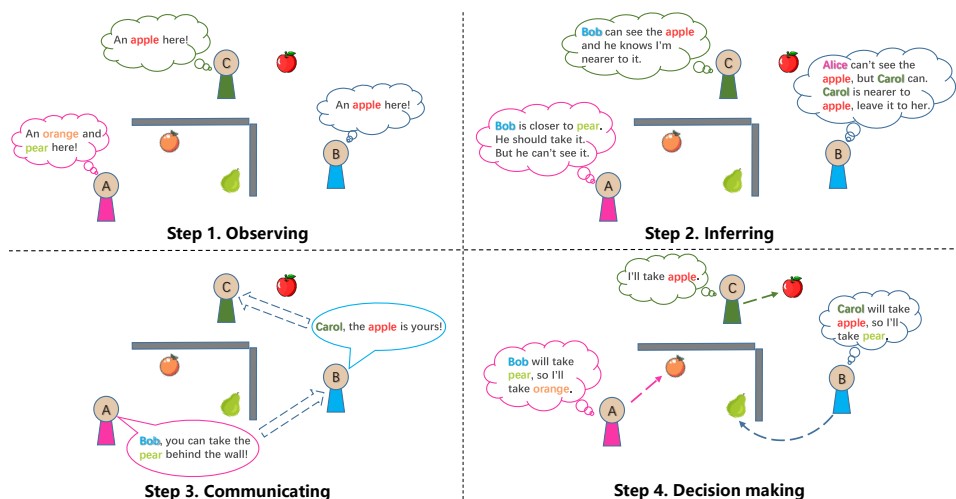

Figure 1: A fruits collection example. The agents are required to cooperatively collect the three target objects (apple, pear, and orange) in the room as fast as possible. The whole process can be divided into 4 steps. In Step 1, 3 agents observe the environment and obtain the state of the visible targets. In Step 2, each agent tries to infer what other agents have seen, and which targets they shall choose as goals. In Step 3, each agent decides whom to communicate with according to the previous inference. In Step 4, each agent decides its own goal of target based on what it observed, inferred, and received.

high-level policy (*planner*) needs to cooperatively choose certain interesting targets as a sub-goal to deal with, such as tracking certain moving objects or navigating to a specific landmark. Then, the low-level policy (*executor*) takes primitive actions to reach the selected goals for $k$ steps. Concretely, each agent receives local observation and estimates the observation of others in the Theory of Mind Network (*ToM Net*). Combining the observed and inferred states, the ToM Net will predict/infer the target choices (intentions) of other agents. After that, each agent decides 'whom' to communicate with according to the local observation filtered by the inferred goals of others. The message is the predicted goals of the message receiver, inferred by the sender. In the end, all the agents decide their own goals by leveraging the observed, inferred, and received information. With the inferring and sharing of intentions, the agents can easily reach a consensus to cooperatively adjust the target-agent relations by taking a sequence of actions.

Furthermore, we also introduce a **communication reduction method** to remove the redundant message passing among agents. Thanks to the Centralized Training and Decentralized Execution (CTDE) paradigm, we measure the effect of the received messages on each agent, by comparing the output of the planner with and without messages. Hence, we can figure out the unnecessary connection among agents. Then we train the connection choice network to cut these dispensable channels in a supervised manner. Eventually, we argue that ToM2C systemically solves the problem of 'when', 'who' and 'what' in multi-agent communication, providing an efficient and interpretable communication protocol for multi-agent cooperation.

The experiments are conducted in two environments. First, in the cooperative navigation scenario (Lowe et al., 2017), the team goal is to occupy landmarks (*static targets*) and avoid collision. Then we evaluate our method in a more complex scenario, multi-sensor multi-target covering scenario (Xu et al., 2020). The team goal of the sensors is to adjust their orientation to cover as many *moving targets* as possible. The results show that our method achieves the best performance (the highest reward and the lowest communication cost) among the state-of-the-art MARL methods, *e.g.,* HiT-MAC (Xu et al., 2020), I2C (Ding et al., 2020), MAPPO (Yu et al., 2021) and TarMAC (Das et al., 2019). Moreover, we further show the good scalability of ToM2C and conduct an ablation study to evaluate the contribution of each key component in ToM2C.

Our contributions are in three-folds: 1) We introduce a cognition-inspired social agent with the ability to infer the mental states of others for enhancing multi-agent cooperation in target-oriented tasks. 2) We provide a ToM-based communication mechanism and a communication reduction method to improve the efficiency of multi-agent communication. 3) We conduct experiments in two typical target-oriented tasks: the cooperative navigation and multi-sensor multi-target coverage problem.

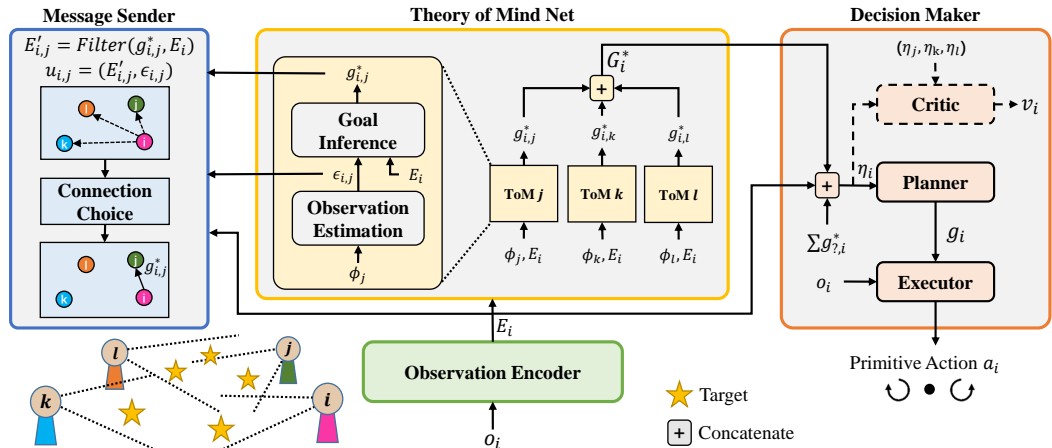

Figure 2: The architecture of ToM2C for each individual. There are four key components: Observation encoder, Theory of Mind net, Message sender, and Decision maker. Each agent first receives a local observation and encodes it with the encoder. Then it performs Theory of Mind inference to estimate the observation of others and predict their goals. Next, it decides 'whom' to communicate with according to local observation filtered by the inferred goals of others. In the end, the planner in decision maker outputs the sub-goal according to what it observes, infers, and receives. The low-level executor takes primitive actions to reach the chosen sub-goal independently.

## 2 RELATED WORK

**Multi-agent Cooperation and Communication.** The cooperation of multiple agents is crucial yet challenging in distributed systems. Agents' policies continue to shift during training, leading to a non-stationary environment and difficulty in model convergence. To mitigate the non-stationarity, the centralized training decentralized execution (CTDE) paradigm is widely employed in the recent multi-agent learning works (Lowe et al., 2017; Foerster et al., 2018; Sunehag et al., 2018; Rashid et al., 2018; Iqbal & Sha, 2019). However, these methods only implicitly guide agents to overfit certain policy patterns of others. Without communication mechanism, agents lack the ability to negotiate with each other and avoid conflicts. As a result, it is hard for the individual agent to quickly adapt to unseen cooperators/environments. Learning to communicate (Sukhbaatar et al., 2016) is a feasible way to promote efficient multi-agent cooperation. Unfortunately, most previous works (Sukhbaatar et al., 2016; Das et al., 2019; Singh et al., 2019) require a broadcast communication channel, leading to huge pressure on bandwidth. Besides, even though I2C (Ding et al., 2020) proposes an individual communication method, the message is just the encoding of observation, which is not only costly but also uninterpretable. (Li et al., 2020a) designs a mechanism to share the camera poses according to the visibility of the target, by exploiting the behavior consistency among agents. In this paper, we will investigate a more efficient peer-to-peer communication, based on theory-of-mind. Hierarchical frameworks (Yang et al., 2019; Kim et al., 2020; Xu et al., 2020) are also investigated to promote multi-agent cooperation/coordination. HiT-MAC (Xu et al., 2020) is the closest work to ours. It proposes a hierarchical multi-agent coordination framework to decompose the target coverage problem into two-level tasks: assigning targets by the centralized coordinator and tracking assigned targets by decentralized executors. The agents in ToM2C are also of a two-level hierarchy. Considering the natural structure of ToMAC, we also decompose the cooperation tasks in this way, rather than learning skills in an unsupervised manner (Yang et al., 2019). Differently, both levels in ToM2C are enabled to perform distributively, thanks to the use of ToM and communication mechanism.

**Theory of Mind.** Theory of Mind is a long-studied concept in cognitive science (Sher et al., 2014; Sanfey et al., 2015; Etel & Slaughter, 2019). However, how to apply the discovery in cognitive science to build cooperative multi-agent systems still remains a challenge. Most previous works make use of Theory of Mind to interpret agent behaviors, but fail to take a step forward to enhance cooperation. For example, Machine Theory of Mind (Rabinowitz et al., 2018) proposes a meta-learning method to learn a ToMnet that predicts the behaviors or characteristics of a single agent. Besides, (Shum et al., 2019) studies how to apply Bayesian inference to understand the behaviors of a group and predict the group structure. (Track et al., 2018) introduces the concept of Satisficing Theory of Mind, which refers to the sufficing and satisfying model of others. (Shu et al., 2021) and (Gandhi et al.,

2021) introduce benchmarks for evaluating machine mental reasoning. These works mainly focus on how to accurately infer the mental state of others, rather than interactive cooperation. (Puig et al., 2021; Carroll et al., 2019; Netanyahu et al., 2021) studies the human-AI cooperation with mental inference. (Lim et al., 2020) considers a 2-player scenario and employs Bayesian Theory of Mind to promote collaboration. Nevertheless, the task is relatively simple and it requires the model of other agents to do the inference. M³RL (Shu & Tian, 2019) proposes to train a manager that infers the minds of agents and assigns sub-tasks to them, which benefits from the centralized mechanism and is not comparable with decentralized methods. (Wu et al., 2021) introduces Bayesian approach for ToM-based cooperation, yet assumes that the environment has a partially ordered set of sub-tasks. Differently, we study how to leverage ToM to guide efficient decentralized communication to further enhance multi-agent cooperation.

**Opponent Modeling.** Opponent modeling (He et al., 2016; Raileanu et al., 2018; Grover et al., 2018; Zhong et al., 2021a) is another kind of method comparable with Theory of Mind. Agents endowed with opponent modeling can explicitly represent the model of others, and therefore plan with awareness of the current status of others. Nevertheless, these methods rely on the access to the observation of others, which means they are not truly decentralized paradigms. Compared with existing methods, ToM2C applies ToM not only to explicitly model intentions and mental states but also to improve the efficiency of communication to further promote cooperation.

## 3 METHODS

In this section, we will explain how to build a target-oriented social agent to achieve efficient multi-agent communication and cooperation. We formulate the target-oriented cooperative task as a Dec-POMDP (Bernstein et al., 2002). The aim of all agents is to maximize the team reward. The overall network architecture is shown in Fig. 2, from the perspective of agent $i$. The model is mainly composed of four functional networks: observation encoder, Theory of Mind network (ToM Net), message sender, and decision maker. First, the raw observation $o_i$, indicating the states of observed targets, will be encoded into $E_i$ by an attention-based encoder. After that, the ToM Net takes Theory of Mind inference $\text{ToM}_i(G_i^*|E_i, \Phi)$ to estimate the joint intention (sub-goals) of others $G_i^*$, according to the encoded observation $E_i$ and the poses of agents $\Phi = (\phi_1, ..., \phi_n)$. In details, taking $ToM_{i,j}$ as an example, it uses the estimated observation $\epsilon_{i,j}$ infers the probability of agent $j$ choosing these targets as its goals, denoted as $g_{i,j}^*$. The estimation of $\epsilon_{i,j}$ is based on the pose $\phi_j$. In general, pose $\phi_j$ indicates the location and rotation of the agent $j$. Specifically, it can be represented as a 6D vector $(x, y, z, roll, yaw, pitch)$ in 3D space and a 3D vector $(x, y, yaw)$ in 2D plane. After the ToM inference, the message sender decides whom to communicate with. Here we employ a graph neural network to model the connection among agents. The node feature of agent $j$ is the concatenation of $\epsilon_{i,j}$ and $E_i$ filtered by $g_{i,j}^*$. The final communication connection is sampled according to the computed graph edge features. Agent $i$ will send $g_{i,j}^*$ to agent $j$ if there exists a communication edge from $i$ to $j$. In the end, we aggregate $G_i^*$, $E_i$ and received messages $\sum g_{?,j}^*$ as $\eta_i$ for the decision making. The planner $\pi_i^H(g_i|\eta_i)$, guided by the team reward, chooses the sub-goal $g_i$ to the low-level executor $\pi_i^L(a_i|o_i, g_i)$, which takes $K$ steps primitive actions to reach the sub-goal.

In the following sections, we will illustrate the key components of ToM2C in detail.

### 3.1 OBSERVATION ENCODER

We employ an attention module (Vaswani et al., 2017) to encode the local observation. There are two prominent advantages of this module. On one hand, it is population-invariant and order-invariant, which is crucial for scalability. On the other hand, multi-target information can be encoded into a single feature due to the weighted sum mechanism. In this paper, we use scaled dot-product self-attention similar to (Xu et al., 2020). The local observation $o_i$ is first transformed to key $K_i$, query $Q_i$ and value $V_i$ through 3 different neural networks. Then the output $E_i = softmax(\frac{\mathbf{Q_i K_i}^T}{\sqrt{d_k}}) \odot \mathbf{V_i}$, where $d_k$ is the dimension of one key. $o_{i,q}$ and $E_{i,q}$ represent the raw and encoded feature of target $q$ to agent $i$ respectively.

### 3.2 THEORY OF MIND NETWORK (TOM NET)

Inspired by the Machine Theory of Mind (Rabinowitz et al., 2018), we introduce the ToM Net that enables agents to infer the observation and intentions of others. For agent $i$, the ToM Net takes the poses of agents $\Phi$ and the encoded local observation $E_i$ as input. Then it infers the observation $\epsilon_i$

and goals $G_i$ of others. Most previous work (He et al., 2016; Raileanu et al., 2018; Lim et al., 2020) consider two-player scenarios, where the agent only needs to model one other agent. Instead, we take a step forward to run our model in a more complex multi-agent scenario consisting of n ($>3$) agents. Therefore, the entire ToM Net of agent $i$ is conceptually composed of $n-1$ separate sub-modules, which focus on modeling the mind of one agent respectively. Shown as Fig. 2, $ToM_i$ is an ensemble of $ToM_{i,j}, ToM_{i,k}, ToM_{i,l},$. In practice, we simply let these models share parameters since all the agents are homogeneous. In this way, we can accordingly adjust the number of sub-modules in ToM Nets with the change of agent number. A single ToM Net is made up of two functional modules: Observation Estimation and Goal Inference.

**Observation Estimation.** The first step of ToM inference is to estimate the observation representation of the other agent. Intuitively, when an agent tries to infer the intention of others, it should first infer which targets are seen by them. Here, we take Bob in Fig. 1 as an example. Before he tries to infer the goals of Alice and Carol, he first infers that Alice cannot observe the apple but Carol can. Similarly, agent $i$ infers the observation of agent $j$, denoted as $\epsilon_{i,j}$, with the pose $\phi_j$. Note that $\epsilon_{i,j}$ is only a representation of the estimated observation. To better learn this representation, we further introduce an auxiliary task, where agents should infer the relation between the targets and other agents, *e.g.,* which fruits are visible or closest to Alice. In practice, we employ the Gated Recurrent Units (Cho et al., 2014) (GRUs) to model the observation of others on time series. In target coverage task, agent $i$ infers which targets are in the observation field of agent $j$ and in cooperative navigation task it infers which landmark is closest to agent $j$.

**Goal Inference.** After agent $i$ finishes the observation estimation of others, it is able to predict which targets will be chosen by them at this step. Just like humans, the agent infers the intentions of others based on what it sees and what it thinks that others see. If we denote this goal inference network as a function GI, then the process can be formulated as $g_{i,j,q}^* = \mathrm{GI}(E_{i,q}, \epsilon_{i,j})$, where $g_{i,j,q}^*$ stands for the probability of agent $j$ choosing target $q$, inferred by $i$. Since there are a total of $n$ agents and $m$ targets in the environment, $G_i^* \in \mathbb{R}_{(n-1)\times m}$. With ToM Net, each agent holds a belief on the observation $\epsilon$ and goal intentions $G^*$ of others. Such belief is not only taken into account for decision making, but also serves as an indispensable component in communication choice.

**Learning ToM Net.** We introduce two classification tasks to learn the ToM Net, which is parameterized by $\theta^{\mathrm{ToM}}$. First, the ToM Net infers the goals $G_i^*$ of others. Note that $g_{i,j,q}^*$ indicates the probability of agent $j$ choosing target $q$, inferred by $i$. Meanwhile, agent $j$ decides its real goals $g_j$. Therefore, $g_j$ can be the label of $g_{i,j}^*$. The Goal Inference loss is the binary cross entropy loss of this classification task:

$$L^{GI} = -\frac{1}{N} \sum_i \sum_{j\neq i} \sum_q [g_{j,q} \cdot \log(g_{i,j,q}^*) + (1 - g_{j,q}) \cdot \log(1 - g_{i,j,q}^*)] \qquad (1)$$

Secondly, the estimated observation $\epsilon$ is additionally guided by the auxiliary task mentioned before. The agent $i$ infers the relation between the targets and agent $j$, denoted as $c_{i,j}^*$. The ground truth is the real observation field $c_j$. $c_{j,q} = 1$ indicates that agent $j$ observes target $q$ or landmark $q$ is closest to agent $j$. Similar to the previous Goal Inference task, this Observation Estimation learning also adopts binary cross entropy loss:

$$L^{OE} = -\frac{1}{N} \sum_i \sum_{j\neq i} \sum_q [c_{j,q} \cdot \log(c_{i,j,q}^*) + (1 - c_{j,q}) \cdot \log(1 - c_{i,j,q}^*)] \qquad (2)$$

$$L(\theta^{\mathrm{ToM}}) = L^{GI} + L^{OE} \qquad (3)$$

### 3.3 MESSAGE SENDER

The message sender leverages the inferred mental state of others from ToM Net to decide 'when' and with 'whom' to communicate, independently. During communication, the message sent from agent $i$ to $j$ is just the inferred intention $g_{i,j}$. Moreover, we further propose a communication reduction method, which can remove useless connections to improve communication efficiency.

In practice, we use Graph Neural Network (GNN), similar to (Li et al., 2020b; Battaglia et al., 2016), to model the communication network in an end-to-end manner. Each agent computes its own graph based on its observation and inferred information of others. Specifically, in the perspective of agent $i$, we make use of the inferred intention $g_i^*$ to filter the observation $E_i$ to generate graph node features. Edge features are computed based on node features, which are further transferred into a probabilistic

distribution over the type of edges(cut or retain). Communication connections are finally sampled according to the probabilistic distribution.

**Inferred-goal Filter.** The feature of agent $j$ in the graph of agent $i$ is the target features filtered by the inferred goals $g_{i,j}^*$ as follows. $\delta$ is a probability threshold, which we set to $0.5$ in this paper. If $g_{i,j,q}^* > \delta$, then agent $i$ will consider target $q$ as the goal that will be chosen by agent $j$. Then we concatenate the filtered feature $E_{i,j}' = \sum_{q=1}^{m} (g_{i,j,q}^* > \delta) \cdot E_{i,q}$ with the estimated observation representation $\epsilon_{i,j}$, to form the estimated node feature $u_{i,j} = (E_{i,j}', \epsilon_{i,j})$. For agent $i$ itself, $u_{i,i} = (\sum_q E_{i,q}, \epsilon_i)$, where $\epsilon_i$ is also computed by Observation Estimation with the pose of $i$.

**Connection Choice.** For a scenario consisting of $n$ agents, there is a total of $n$ directed graphs $\mathcal{G} = (\mathcal{G}_1, \mathcal{G}_2, ... \mathcal{G}_n)$. $\mathcal{G}_i = (\mathcal{V}_i, \mathcal{E}_i)$ is the local graph for agent $i$ to compute the communication connection from agent $i$. The vertices $\mathcal{V}_i = \{f(u_{i,j})\}$, where $f$ is a node feature encoder. Edges $\mathcal{E}_i = \{\sigma(u_{i,j}, u_{i,k})\}$, where $\sigma$ is an edge feature encoder. Like the Interaction Networks (IN) (Battaglia et al., 2016), we propagate the node and edge features spatially to obtain node and edge effects. For convenience, we will describe only graph $\mathcal{G}_i$ in the following formula and omit the index $i$. Let $V_j$ be the encoded node feature of $j$, and $h_j$ be the node effect. Similarly, let $\mathcal{E}_{j,k}$ be the encoded edge feature, $h_{j,k}$ be the edge effect. Initially, $h_j = V_j, h_{j,k} = \mathcal{E}_{j,k}$. Then the graph iterates several times to propagate the effect:

$$h_j = \Psi^{\text{node}}(V_j, h_j, \sum_k h_{k,j}) \tag{4}$$

$$h_{j,k} = \Psi^{\text{edge}}(h_j, h_k, h_{j,k}) \tag{5}$$

In the end, we obtain the final edge feature $(\mathcal{E}_{i,j}, h_{i,j})$, and compute the probabilistic distribution over the type of the edge $(p_{\text{cut}}^{i,j} + p_{\text{retain}}^{i,j} = 1)$. Here we apply the Gumbel-Softmax trick (Jang et al., 2016; Maddison et al., 2016) to sample the discrete edge type, so the gradients can be back-propagated in end-to-end training. Considering that it is the local communication graph of agent $i$, only the types of $\mathcal{E}_{i,-i}$ are sampled. If edge $\mathcal{E}_{i,j}$ is retained, agent $i$ will send $g_{i,j}^*$ to $j$.

**Communication Reduction (CR).** In practice, the GNN tends to learn a densely connected graph, even if applying a sparsity regularization while learning. However, we observe that the decisions made by receivers are not always influenced by the received messages, indicating that some connections are actually redundant in the GNN. Therefore, it is necessary for us to figure out the really valuable connections from the densely connected networks. To this end, we measure the effectiveness of each connection by taking an ablative analysis. To be specific, we observe the non-communicative sub-goals $g_i^-$ of agent $i$ by removing the received message in $\eta_i$. We can estimate the effect of the received messages to agent $i$ by measuring the KL-divergence, referred as $\chi = D_{KL}(g_i^- || g_i)$. Here we set a constant threshold $\tau$ to generate a binary pseudo label for learning to remove redundant connections. Specifically, if $\chi < \tau$, we regard that the messages are redundant to agent $i$. Thus the edges pointing at $i$ will be labeled as 'cut', $l_{*,i} = 0$. Otherwise ($\chi > \tau$), labeled as 'retain', $l_{*,i} = 1$. Then the tuning of message-sender network follows the binary cross entropy loss:

$$L^{CR} = -\frac{1}{M} \sum_i \sum_j [l_{i,j} \cdot \log(p_{\text{retain}}^{i,j}) + (1 - l_{i,j}) \cdot \log(p_{\text{cut}}^{i,j})] \tag{6}$$

## 3.4 DECISION MAKER

Once the agent receives all the messages, it can decide its own sub-goals of targets based on its observation, inferred intentions of others and received messages. Therefore, the actor feature $\eta_i = (E_i, \max_j g_{i,j}^*, \sum_s g_{s,i}^*)$ is the input to the actor network. The second term $\max_j g_{i,j}^*$ refers to the max inferred probability of a target to be chosen by another agent. The third term $\sum_s g_{s,i}^*$ refers to the sum of the messages from others, indicating how much certain others infer that agent $i$ should choose the target. The actor decides its goals $g_i$ according to $\eta_i$. The centralized critic obtain global feature $(\eta_1, ... \eta_n)$ to compute value. The low-level executor $\pi_i^L(a_i | o_i, g_i)$ takes primitive actions to accomplish the sub-goal. Similar to HiT-MAC (Xu et al., 2020), the high-level planner is guided by the team reward. The low-level executor is guided by the goal-conditioned reward, measuring the quality of the achievement of the sub-goals.

## 3.5 TRAINING

Following the Centralized Training Decentralized Execution (CTDE) paradigm (Rashid et al., 2018), we also train our model in a centralized manner, *i.e.,* feeding the global observation to the critic and

using the actual observation and goals of all agents to supervise the ToM Net. The overall network is trained by Reinforcement Learning (RL) in an end-to-end manner. We adopt standard A2C (Mnih et al., 2016) as the RL algorithm, while any MARL method with CTDE framework is also applicable, such as PPO (Schulman et al., 2017; Yu et al., 2021). Besides, as we mentioned in Sec. 3.2, we provide the actual observation and goals of other agents to supervise the training of the ToM net. We also apply communication reduction(sec. 3.3) to tune the Message-Sender network.

**Training Strategy.** We find that it is hard for an agent to learn long-term planning from scratch. Therefore, we set the initial episode length $L$ and discount factor $\gamma$ to a low value, forcing agents to learn short-term planning first. During training, the episode length and discount factor $\gamma$ increase gradually, leading the agents to estimate the value on a longer horizon. Furthermore, we freeze the ToM net while the other parts of the model (parameterized by $\theta^{\text{other}}$) are updated through RL. The reason is that the ToM net infers the goals of others, and the policy network is continuously updated during RL training. Meanwhile, the output of ToM net is a part of the input to the policy network. If we train them simultaneously, they are likely to influence each other in a nested loop. Therefore, we only collect the ToM inferred data into a batch during RL training. Once the batch is large enough, we stop RL and start ToM training to minimize ToM loss in Eq. 3. More details about the model and training strategy can be found in Appendix. B.

## 4 EXPERIMENTS

We evaluate ToM2C in two typical target-oriented multi-agent tasks: cooperative navigation (CN) (Lowe et al., 2017) and multi-sensor multi-target coverage (MSMTC) (Xu et al., 2020). CN is the simplest environment, where targets (landmarks) are *static* and the agent only needs to choose *one target* as sub-goal. In CN, $n$ agents need to occupy $n$ landmarks (targets) so as to minimize the distances of all landmarks to their nearest agents. All agents share a team reward, which is the sum of the negative distance of each landmark to its nearest agent. If two agents collide, the team will get a penalty -1. In MSMTC, $n$ agents need to cooperatively adjust the sensing orientation to maximize the coverage rate of $m$ moving targets. As is shown in Fig.3, each sensor can only see the targets that are within the radius and not blocked by any obstacle. The reward is the coverage rate of targets. If there is no target covered by sensors, we punish the team with a reward $r = -0.1$. For MSMTC, all targets are *dynamic* and each agent chooses *multiple targets* concurrently as its sub-goals. Therefore, CN and MSMTC respectively represent two typical settings of ToMAC: *static target & one-choice sub-goal* and *dynamic targets & multi-choice sub-goal*. More details can be found in Appendix. A.

In the following, we compare ToM2C with baselines in the two scenarios. Then, in MSMTC, we conduct an ablation study, analyze the communication, and evaluate the scalability of ToM2C.

### 4.1 COMPARE WITH BASELINES

We compare our methods with 4 baselines. TarMAC (Das et al., 2019) is a multi-agent communication method that requires a broadcast channel. I2C (Ding et al., 2020) proposes an individual communication mechanism, which is also achieved by ToM2C. HiT-MAC (Xu et al., 2020) is a hierarchical method that uses a centralized coordinator to organize the agents. MAPPO (Yu et al., 2021) is a variant of PPO which is specialized for multi-agent settings, running without communication. we also implement a global heuristic search algorithm in MSMTC , as a reference. This policy searches in one step for the primitive actions

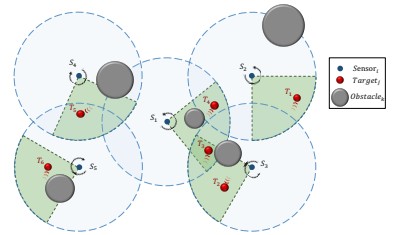

Figure 3: An example of the target coverage environment with obstacles.

of all the sensors to minimize the sum of minimum angle distance of a target to a sensor. Note that the heuristic search policy exploits the global state to do the global search, while all the other methods are restricted to local observation. Therefore the heuristic search policy can serve as a reference 'upper bound' that evaluates all the MARL baselines. Please refer to Appendix. C for more details.

ToM2C is a hierarchical method, while some of the baselines are not. In MSMTC, we let all of the methods share the low-level rule-based policy adopted in ToM2C. Therefore, these baselines are only used for training the high-level policy, same as ToM2C. In CN, ToM2C and HiT-MAC share the low-level policy and other methods keep non-hierarchical.

**Quantitative Results.** In CN, as Tab. 1 shows, ToM2C and HiT-MAC are far better than other methods. Here we also add MADDPG as a baseline method, because this environment is early adopted in the original paper of MADDPG (Lowe et al., 2017). The performance of HiT-MAC is

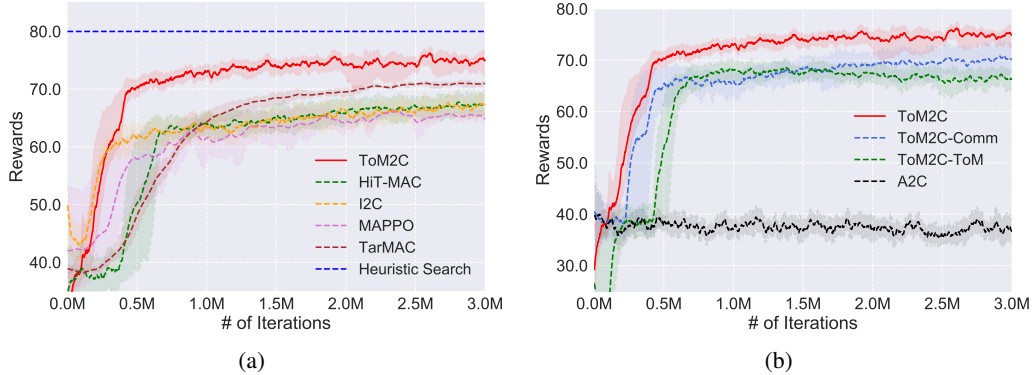

(a)                                           (b)

Figure 4: The learning curve of our method with baselines and reference policies in the MSMTC scenario. The learning-based methods are all trained in the environment with 4 sensors and 5 targets. (a) comparing ours with baselines; (b) comparing ours with its ablations.

Table 1: Quantitative Results in Cooperative Navigation ($n = 7$).

|  | ToM2C | TarMAC | I2C | HiT-MAC | MADDPG | MAPPO |
|---|---|---|---|---|---|---|
| Reward | -0.79 ± 0.39 | -2.14 ± 0.24 | -1.59 ± 0.40 | -0.61 ± 0.14 | -2.75 ± 0.61 | -2.47 ± 0.21 |

slightly better than ToM2C. It is because the hierarchical structures of ToM2C and HiT-MAC are different. For HiT-MAC, there is a centralized coordinator that collects all the local observations and assigns the goals for each agent. On the contrary, ToM2C agents run in a fully decentralized manner. Agents only have access to local observations and decide their own goals. The hierarchy structure is only used for separate goal selections and primitive actions. Due to the centralized manner, HiT-MAC is likely to degrade in more complex scenarios (*e.g.,* large-scale MSMTC). Moreover, HiT-MAC is also more costly in communication (Sec. 4.3) and weaker in scalability (Sec. 4.4). Therefore, ToM2C is not inferior to HiT-MAC.

In MSMTC, as Fig.4(a) shows, ToM2C achieves the second highest reward (75) in the setting of 4 sensors and 5 targets, only lower than the searching policy (80). The reward performance of I2C, MAPPO and HiT-MAC are all around 66. TarMAC reaches a reward of 71, which is superior to the other three baselines. This could be attributed to the broadcast channel adopted by TarMAC. Since all the methods leverage the hierarchy structure, the superiority of ToM2C comes from the model and training strategies.

## 4.2 ABLATION STUDY

We conduct this study to evaluate the contribution of two key components in our model: ToM net and Message sender. The ToM2C-Comm model abandons communication, so the actor makes decisions only based on local observation and inferred goals of others. The ToM2C-ToM abandons ToM net, but keeps the Messages sender. However, as explained before, the local graph node feature is computed based on the ToM net output. To deal with this problem, we use the encoded observation $E_j$ to replace the original node feature $u_{i,j}$. In this way, the $n$ local graphs degrade into one global graph, so the ToM2C-ToM model actually breaks the local communication mechanism. If we remove both the ToM net and Message sender, ToM2C will become pure A2C. We show in Fig.4(b) that if we abandon one of the key components, the performance will drop. Considering that ToM2C-Comm outperforms ToM2C-ToM and ToM net is actually essential for communication, we argue that ToM net mainly contributes to our method.

## 4.3 COMMUNICATION ANALYSIS

We compare our method with several candidates in regard to communication expenses. There are 2 metrics here: the number of communication edges and communication bandwidth. The latter metric considers both the count of edges and the length of a single message. TarMAC utilizes a broadcast channel, so there are two communication edges between each pair. The communication in HiT-MAC is between the executors and the coordinator. ToM2C w/o CR refers to the ToM2C model without communication reduction. Apparently, communication bandwidth is more significant in real applications, so we place the figure of communication bandwidth in the main text.

The statistics of communication edges can be found in Appendix. D. As is shown in Fig.5, the communication bandwidth of ToM2C and ToM2C without CR are much lower than TarMAC, I2C and HiT-MAC. It is because in ToM2C the message is only the inferred goals, while TarMAC, I2C and HiT-MAC have to send the local observation. Therefore, the single message in ToM2C is much simpler than that of TarMAC, I2C and HiT-MAC. As a result, the communication cost of ToM2C is extremely less than existing methods.

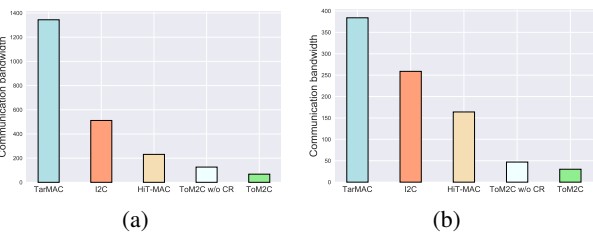

Figure 5: The communication bandwidth of models in (a) Cooperative navigation and (b) MSMTC.

## 4.4 SCALABILITY

We evaluate the scalability of our method to a different number of sensors and targets in the MSMTC task. Note that the model is only trained in the setting of 4 sensors and 5 targets, so this could be regarded as zero-shot transfer. Since the difficulty of the task changes with the scale of cameras and targets, we make use of the heuristic search policy, whose target coverage rate in each setting can serve as a referential value. We adopt such

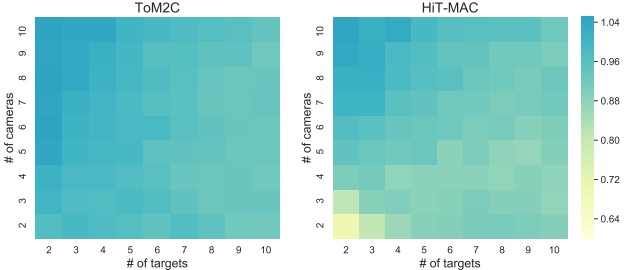

Figure 6: Analyzing the scalability in scenarios with different sizes of cameras and targets. The left heatmap shows $R^{\mathrm{ToM}}$. The right heatmap shows $R^{\mathrm{HM}}$.

a metric that divides the coverage rate of the model by the coverage rate of the search policy: $R^{\mathrm{ToM}}_{p,q} = C^{\mathrm{ToM}}_{p,q}/C^{\mathrm{HS}}_{p,q}$, where $C^{\mathrm{ToM}}_{p,q}$ refers to the coverage rate of ToM2C in the setting of $p$ cameras and $q$ targets and $C^{\mathrm{HS}}_{p,q}$ refers to the coverage rate of heuristic search. In this way, we may reduce the fluctuation induced by the change of task difficulty. To better show the advantage of our method, we further test the scalability of HiT-MAC. The experiments are conducted in a total of 81 settings, where cameras and targets both range from 2 to 10. We compute $R^{\mathrm{ToM}}$ and $R^{\mathrm{HM}}$(ratio of HiT-MAC and heuristic search) in all the settings and draw the heatmap in Fig. 6. It is shown in the left figure that the ratio $R^{\mathrm{ToM}}$ stays near 1, which means that the performance of ToM2C is close to heuristic search in all the settings. Furthermore, if we compare the heatmap of ToM2C and HiT-MAC, it is obvious that ToM2C is more stable than HiT-MAC when transferred to different scales. In this way, we show that ToM2C has a rather strong generalization.

## 5 CONCLUSION AND DISCUSSION

In this work, we study the target-oriented multi-agent cooperation (ToMAC) problem. Inspired by the cognitive study in Theory of Mind (ToM), we propose an effective Target-orient Multi-agent Cooperation and Communication mechanism (ToM2C) for ToMAC. For each agent, ToM2C is composed of an observation encoder, ToM net, message sender, and decision maker. The ToM net is designed for estimating the observation and inferring the goals (intentions) of others. It is also deeply used by the message sender and decision maker. Besides, a communication reduction method is proposed to further improve the efficiency of the communication. Empirical results demonstrated that our method can deal with challenging cases and outperform the state-of-the-art MARL methods.

Although impressive improvements have been achieved, there are still some limitations of this work leaving for addressed by future works. 1) It is necessary to further evaluate the model on other applications. As each component in the model is general, we are confident to apply ToM2C to other target-oriented tasks (e.g. SMAC (Samvelyan et al., 2019)) in the future. It is also interesting to extend ToM2C to non-target environments (e.g. Hanabi (Bard et al., 2020)), which requires a further definition of sub-goals or automatic goal generation. 2) Besides, the communication reduction method can also be further optimized, as the pseudo labels we generated for communication reduction are noisy in some cases.

## REPRODUCIBILITY STATEMENT

The code is available at https://github.com/UnrealTracking/ToM2C. The details of the environments can be found in Appendix. A. The hyper-parameters used in ToM2C are listed in Tab. 2.

## ACKNOWLEDGEMENTS

This work was supported by MOST-2018AAA0102004, NSFC-62061136001, China National Post-doctoral Program for Innovative Talents (Grant No. BX2021008), Qualcomm University Research Grant.

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

# A    DETAILS OF ENVIRONMENT

## A.1    MULTI-SENSOR TARGET COVERAGE

The multi-sensor multi-target tracking environment is developed based on the previous version in HiT-MAC(Xu et al., 2020), and it inherits most of the characters. It is a 2D environment that simulates the real target coverage problem in directional sensor networks. Each sensor can only see the targets that are within the radius and not blocked by any obstacle. There 2 types of target: destination-navigation and random walking. The former one moves in the shortest path to reach a previously sampled destination. The latter one moves randomly at each time step. At the beginning of each episode, the location of sensors, targets and obstacles are randomly sampled. Besides, the types of the targets are also sampled according to a pre-defined probability. The length of an episode is 100 steps.

**Observation Space.**    At each time step, the local observation $o_i$ is a set of agent-target pairs: $(o_{i,1}, ... o_{i,m})$. If target $q$ is visible to agent $i$, then $o_{i,q} = (i, q, d_{i,q}, \alpha_{i,q})$, where $d_{i,q}$ is the distance and $\alpha_{i,q}$ is the relative angle. If target $q$ is not visible to $i$, then $o_{i,q} = (0, 0, 0, 0)$. Therefore, $o_i \in \mathbb{R}_{m \times 4}$.

**Action Space.** The primitive action for a sensor is to stay or rotate +5/-5 degrees. For our method, the high-level action is the chosen goals $g_i$, which is a binary vector of length m. $g_{i,q} = 1$ means the agent chooses target $q$ as one of its goals. $g_{i,q} = 0$ means not. Although the low-level executor can be trained by reinforcement learning (RL) as HiT-MAC, we find a simple rule-based policy can also work well in most cases. Therefore we only train the high-level policy.

**Reward.** Reward is the coverage rate of targets: $r = \frac{1}{m} \sum_q I_q$, where $I_q = 1$ if $q$ is covered by any sensor. If there is no target covered by sensors, we punish the team with a reward $r = -0.1$.

Compared with the version in HiT-MAC, we add some new features to make it more complex and realistic:

- First, HiT-MAC assumes that all the agents can see all the targets in the environment, which is unreasonable for real applications. Therefore, we change this setting into a Dec-POMDP, in which an agent can only obtain the information in its local observation. For those targets that are out of view, the corresponding observation will be a zero vector.

- Secondly, we add another kind of objects, obstacles, into the environment. The obstacles are all circles in this 2D plain simulator, varying in the radius. The targets within the observation radius will still be invisible if it is shadowed by an obstacle.

- Finally, in the original environment all the targets move in a goal-oriented manner. The targets sample their destinations at the beginning of an episode and navigate themselves to the destinations. Nevertheless, not all targets in real world follow the same action pattern. Therefore, we fill the environment with a mixed-type population of targets. The target can either be goal-oriented or random-walking. When the target is random-walking, it will randomly sample a primitive action to take at each step. In this way, the movement of targets is harder to predict, raising the difficulty of the planning.

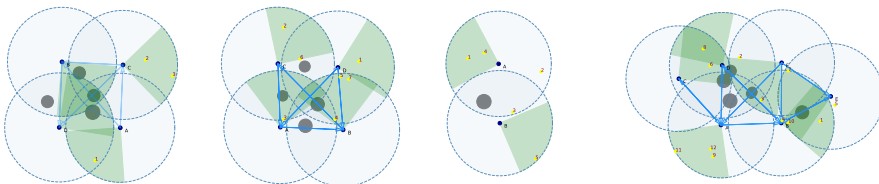

Figure 7: Snapshots of the multi-sensor multi-target tracking environments with different scales. From the upper left figure, in the clockwise direction, they are 4 vs 3, 4 vs 7, 2 vs 5, and 6 vs 12, respectively.

## A.2 COOPERATIVE NAVIGATION

The environment is similar to the one used in MADDPG (Lowe et al., 2017). $n$ agents need to cooperatively reach $n$ landmarks. In this task, we set $n = 7$. The length of each episode is 100 steps. For hierarchical methods, the high-level episode length is set to $K = 10$. Thus, each high-level step consists of 10 low-level steps.

**Observation Space.** At each time step, the observation $o_i$ is the concatenation of self location, self velocity, location and velocity of visible agents, location of visible landmarks.

**Action Space.** The primitive action for an agent is to move up/down/left/right. For our method, the high level action is the chosen goal $g_i$. Similar to the multi-sensor target coverage task, ToM2C and HiT-MAC again make use of a rule-based low-level policy.

**Reward.** The team reward is the sum of the negative distance of each landmark to its nearest agent. If two agents collide, the team will get a penalty $r = -1$.

## B ToM2C DETAILS

**Network Architecture and Hyper-parameters for ToM2C.** The observation encoder consists of 2-layer multilayer perceptron (MLP) and an attention module: $\text{att}_1$. The ToM net consists of a Gated Recurrent Unit (GRU) and 2-layer MLP. The message sender is a Graph Neural Network (GNN) and the actor consists of one fully connected layer. The critic consists of an attention module $\text{att}_2$ that can handle different numbers of agents. As mentioned before, the basic RL training algorithm is A2C, and the hyper-parameters are detailed in Tab. 2.

Table 2: Hyper-parameters for ToM2C

| Hyper-parameters | # | Description |
|---|---|---|
| GRU hidden units | 32 | the # of hidden units for GRU |
| $\text{att}_1$ hidden units | 64 | the # of hidden units for $\text{att}_1$ |
| $\text{att}_2$ hidden units | 192 | the # of hidden units for $\text{att}_1$ |
| max steps | 3M | maximum environment steps sampled in workers |
| episode length | 100 | maximum time steps per episode |
| discount factor | 0.9 | discount factor for rewards |
| entropy weight | 0.005 | parameter for entropy regularization |
| learning rate | 1e-3 | learning rate for all networks |
| workers | 6 | the # of workers for sampling |
| update frequency | 20 | the network updates every # steps in A2C |
| ToM Frozen | 5 | the ToM net is frozen for every # times of RL training |
| gamma rate | 0.002 | the increasing rate of discounting factor $\gamma$ |

**Training Strategy.** There are two training strategies adopted to accelerate training and stabilize the result. As mentioned in Sec.3.5, one is to increase episode length $L$ and $\gamma$ factor gradually during training, the other one is to split the optimization of the ToM and RL model.

In this paper, we propose this curriculum learning strategy that gradually increases episode length $L$ and discounting factor $\gamma$. Usually, the discounting factor $\gamma$ is set larger than 0.9 to encourage long-term planning in RL algorithms. Furthermore, the length of an episode is usually determined by the environment. We notice that if using the default hyper-parameters, the agents are sample inefficient and unstable while learning. In our experiments, we set $L = 20$ and $\gamma = 0.1$ initially. After 2000 episodes of warm-up, the $\gamma$ factor will be updated according to a pre-set rate $\beta$. Each time the network is optimized through reinforcement learning, $\gamma = \gamma * (1 + \beta)$, where $\beta = 0.002$ in this paper. Simultaneously, the episode length $L$ is updated with $\gamma$. In fact, $L = \lfloor \frac{\gamma + 0.1}{0.2} \rfloor \times 20$. In the end, $\gamma = 0.9$ and $L = 100$. By doing so, the agents learn short-term planning first, and then adapt to a longer horizon. We find in experiments that such strategy accelerates the training process, leading to a faster convergence and a better performance.

Furthermore, we separate the optimizations of the ToM and RL model in implementation. Before the training process starts, the parameters of our model are split into two parts: $\theta^{ToM}$ and $\theta^{other}$. Each part is optimized individually by a different optimizer. Since we adopt A2C as the base RL training algorithm, we collect trajectories data from different worker processes and send them to the training process when all the running episodes end. After that, $\theta^{other}$ is optimized with regard to the A2C loss. Meanwhile, the trajectories data for ToM training are saved instead of being used for training ToM net immediately. In this way, the ToM net is 'frozen'. $\theta^{ToM}$ will be optimized with regard to ToM loss after $\theta^{other}$ has been optimized for $T_F$ times. Here we choose $T_F = 5$. Just like the discussion before, the separation of ToM and RL training avoids the nested loop of influence among the ToM net and the policy network.

The environment and model are implemented in Python. The model is built on PyTorch and is trained on a machine with 7 Nvidia GPUs (Titan Xp) and 72 Intel CPU Cores.

## C  BASELINES

**Heuristic Search Method.**  To evaluate the performance of our ToM2C model, we choose to implement a heuristic search policy to serve as a reference. This search policy is applied to select low-level sensor action(Stay, Turn Left/Right). At each step, the policy searches all the $3^n$ possibilities of combination of actions, where $n$ is the number of sensors. The goal is to find the action combination that minimizes the angle distance of targets to sensors. Specifically, we denote the angle distance of target $j$ to sensor $i$ as $\alpha_{ij}$. Then the objective is to minimize $\sum_{j=1}^{m} \min_i \{\alpha_{ij}\}$. It is obvious that such searching policy only considers one step, thus not the optimal policy. However, we show that this naive heuristic search can reach 80% target coverage. As a result, it can serve as a reference 'upper bound' that evaluates all the MARL baselines.

**MARL Baselines.** The code of HiT-MAC and I2C are from their official repositories. We follow the default hyper-parameters in their code, except that we change the learning rate, discounting factor $\gamma$ and episode length to be the same as ToM2C. TarMAC is implemented on our own because no official code is released. Moreover, HiT-MAC is a hierarchical method, so we simply train the high-level coordinator and use the same rule-based low-level policy utilized in ToM2C. On the other hand, I2C is not a hierarchical method and it is not target-oriented. As a result, we concatenate all the target information into one vector as the observation for I2C. The action space is modified as the set of choice of all the targets, so the space size is $2^m$, where $m$ is the number of targets. In this way, the output action of I2C agent is the selection of goal targets, same as HiT-MAC and ToM2C. Once the goal target is selected, the primitive actions will be chosen by the rule-based policy.

## D  QUANTITATIVE RESULTS

For MSMTC task, We list the **coverage rate** achieved by different methods in Tab. 3. The mean and standard deviation are computed based on the data collected in 1000 episodes. The performance in cooperative navigation is listed in Tab. 1.

Table 3: Coverage Rate in 4 sensors vs 5 targets scenario

| Methods | Coverage Rate(%)↑ |
|---|---|
| A2C | $38.44 \pm 0.54$ |
| HiT-MAC | $61.48 \pm 1.45$ |
| I2C | $66.29 \pm 1.40$ |
| MAPPO | $66.87 \pm 0.69$ |
| ToM2C-ToM | $67.66 \pm 0.63$ |
| TarMAC | $70.56 \pm 0.81$ |
| ToM2C-Comm | $71.61 \pm 0.31$ |
| **ToM2C(Ours)** | $\mathbf{75.38 \pm 0.57}$ |

Apart from the coverage rate, we analyze the communication efficiency of different methods. There are 2 metrics introduced in this paper. Communication edges refer to the count of directed communi-

Table 4: Communication Statistics of MSMTC

| Methods | Communication Edges | Communication Bandwidth↓ |
|---|---|---|
| TarMAC | $12.00 \pm 0.00$ | $384.00 \pm 0.00$ |
| I2C | $7.19 \pm 0.13$ | $258.84 \pm 4.16$ |
| HiT-MAC | $8.00 \pm 0.00$ | $164.00 \pm 0.00$ |
| ToM2C w/o CR | $9.39 \pm 0.19$ | $46.93 \pm 0.97$ |
| ToM2C(Ours) | $\mathbf{6.03 \pm 0.20}$ | $\mathbf{30.15 \pm 1.02}$ |

Table 5: Communication Statistics of CN

| Methods | Communication Edges | Communication Bandwidth↓ |
|---|---|---|
| TarMAC | $42.00 \pm 0.00$ | $1344.00 \pm 0.00$ |
| I2C | $14.21 \pm 1.65$ | $511.41 \pm 59.44$ |
| HiT-MAC | $14.00 \pm 0.00$ | $231.00 \pm 0.00$ |
| ToM2C w/o CR | $42.00 \pm 0.00$ | $126.00 \pm 0.00$ |
| ToM2C(Ours) | $\mathbf{22.48 \pm 0.91}$ | $\mathbf{67.44 \pm 2.74}$ |

cation pairs. One edge from $i$ to $j$ means that agent $i$ sends a message to agent $j$. Communication bandwidth refers to the total volume of messages. As we mentioned in the experiment section, it is the volume of messages that has a decisive effect on the cost of communication. Since the messages are all float-type vectors, we use the length of a message instead of the number of bits to represent the volume of a single message. For I2C, the message from agent $i$ is the local observation $o_i$, containing the information of all the targets. For HiT-MAC, communication happens between the executors and the coordinator. The executors send their local observation to the coordinator, and the coordinator returns the goal assignment. For ToM2C w/o CR and ToM2C, the message is simply the inferred goals of the receiver. ToM2C w/o CR means that the trained ToM2C model is not further optimized to reduce communication.

The experiments are conducted in the 4-sensors-and-5-targets scenario and 7v7 cooperative navigation. As shown in Tab. 4 and Tab. 5, our method achieves the lowest communication cost.

**ToM Accuracy.** We further tested the accuracy of ToM prediction, including goal inference and observation estimation. In MSMTC, the accuracy of goal inference is $80.2\% \pm 1.4\%$. The accuracy of observation estimation is $98.1\% \pm 0.3\%$. For comparison, the accuracy of random prediction are $50.1\% \pm 0.3\%$ and $50.4\% \pm 0.6\%$ respectively.

**Pose Accessibility.** In this paper, we enable the agents to have access to the poses of all the other agents, while restricting the observability of targets. To further validate the performance of ToM2C in the case of restricted pose access, we add an observation distance between agents. If agent $j$ is out of the view of agent $i$, we mask the ToM inference and communication to agent $j$ from agent $i$. Under such partially observable agent poses, we test our model in MSMTC. The average rewards among 100 episodes are listed in Tab. 6, comparing with the case of using all agent poses. It is shown that the performance of ToM2C is still comparable to the original version (use all agent poses) when the poses of agents are partially observable.

Table 6: Comparison between partially and fully observable poses in MSMTC.

| sensors vs. targets | Partially observable agent poses | All agent poses |
|---|---|---|
| 10 vs. 10 | $75.96 \pm 0.81$ | $77.26 \pm 0.52$ |
| 10 vs. 5 | $85.06 \pm 0.32$ | $84.69 \pm 0.77$ |
| 4 vs. 5 | $74.73 \pm 0.95$ | $75.38 \pm 0.57$ |

# E    DEMO SEQUENCE

To better understand the learned behavior, we render the target coverage environment and show a typical demo sequence in Fig. 8. It consists of 4 consecutive keyframes in one episode. The arrows between sensors indicate communication connections. Note that communication only happens every 10 steps. In step 16, sensor $D$ can track target 1, 2 and 4. However when it comes to step 22, sensor $D$ can no longer track all the three targets, so it starts to hesitate about which targets to track. Then in step 24, $A$ sends a message to $D$, and $D$ inferred that $A$ would track target 1 and 2. Therefore, it re-plans its own goal to be target 4. In the end, we can see that sensor $D$ really abandons target 1 and 2, and focuses on target 4.

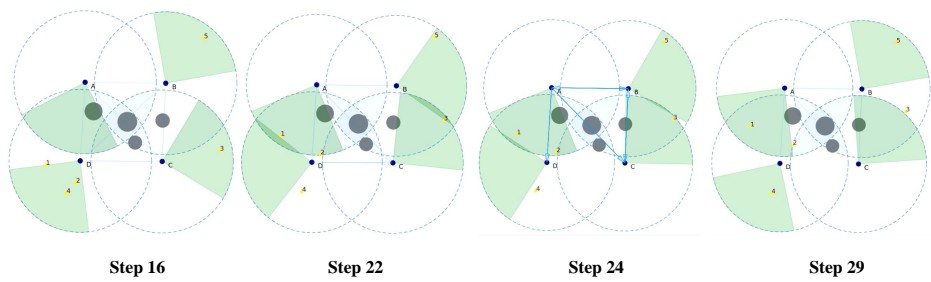

|  Step 16 | Step 22 | Step 24 | Step 29 |

Figure 8: An exemplar sequence in 4 sensors and 5 targets MSMTC environment. The gray circle indicates the obstacle. The arrows are rendered as solid only when the communication happens, and transparent at other times.

