# OpenReview forum: "ToM2C: Target-oriented Multi-agent Communication and Cooperation with Theory of Mind"
_ICLR.cc/2022/Conference — ICLR 2022 Poster_

### Official Review · Reviewer_V4LB · 2021-10-28

**Correctness:** 4
**Technical Novelty And Significance:** 3
**Empirical Novelty And Significance:** 3
**Recommendation:** 6
**Confidence:** 4

**Main Review:**

I commend the authors for designing and implementing this highly complex agent architecture, which clearly exceeds the complexity of many previous multi-agent algorithms (having seen the supplementary material, I see the agent model is thousands of lines of code). Precisely because of the complexity, it is imperative that the technical writing is organized perfectly to ensure that readers appreciate the work and are not put off due to the complexity. In its present form, this paper hasn't met that standard. I believe the architecture is a valuable contribution to the advancement of agent design, especially for cooperative MARL, so I encourage the authors to take the following criticism in a positive way to improve the writing.

1. Referring to the abstract, do agents really "reach a consensus"? By definition, a consensus is achieved when many agents all converge to the same decision or estimation of a quantity. This paper is concerned with multi-goal problems, so clearly agents don't all choose the same goal. If the authors intend to say that agent $i$'s prediction of agent $j$'s goal matches agent $j$'s actual chosen goal, then they authors need to verify this in their experiments (I don't see such a test) and describe it as so. Calling that "consensus" is confusing.
2. Still in the abstract, authors say agents chooose "sub-goals". Does this mean an agent chooses many different sub-goals at various steps in an episode? How many primitive actions are taken per sub-goal? Is each sub-goals drawn from a predefined set of possible goals? Or is a sub-goal a latent variable whose meaning emerges via some kind of unsupervised learning? Or are there hand-designed rewards for each sub-goal? I have these questions even after multiple readings of the paper. Reference [1] below is an example of such details that one expects from hierarchical MARL methods.
3. Page 3, related work section, the authors claim that CTDE methods are flawed becaue "coooperation collapse in spite of a slight change in the team formation,..." Which work has shown this? What is meant by "change in team formation"? Plenty of works have shown excellent generalization, especially since many multi-agent environments have localized interactions, so centralized training with few agents is sufficient for generalizing to decentralized settings with many more agents.
4. Page 4, "pose" $(\phi_1,\dotsc, \phi_n)$ is undefined. It also seems way too specific to particular applications.
5. Page 4, "The final communication connection is sampled according to the computed graph edge features." How are edge features defined? Even after reading to the end of the methods section, I still can't find the answer.
6. Section 3.2, authors say "Therefore, the entire ToM net of agent i is actually composed of n-1 separate ToM nets", but then later say it's actually just a single model because agents are homogeneous and one model can be used to process all other agents' info. Be consistent.
7. Each agent infers the intended goal of other agents. The accuracy of this prediction needs to be shown in an experiment.
8. Page 4, last line, authors mention an "auxiliary task" for agent $i$ to estimate the observation of agent $j$. It turns out, two pages later in page 6, that this auxiliary task is that agent $i$ predicts whether agent $j$ can see a landmark. Authors should define that auxiliary task immediately after introducing it, not 2 pages later. Also, the accuracy of this prediction needs to be shown in an experiment.
9. Page 5, in the paragraph "connection choice", what is an "effect", as in "node effect" and "edge effect"? This is not a standard term in the graph neural network literature.
10. Page 6, in the "Communication Reduction (CR)" paragraph, there's a model that trims the edges of the communication graph, and it is "supervised by the generated pseudo labels". What exactly is this thing? What are the inputs and outputs? Is it a binary classifier that decides whether to cut or not cut an edge? This is completely unclear.
11. Section 3.4, first paragraph, what do authors mean by an "actor-critic feature"? The observation input to the policy network?
12. Section 3.4, first paragraph, authors say "The actor decides its goals $g_i$ according to $\eta_i$. How does actor "decide $g_i$ according to $\eta_i$"? What is the reward for the goal-selector? What is the reward for the low-level executor? By goal $g_i$ and "sub-goal", do you mean the same thing? What is the reward corresponding to each sub-goal?

Of course, some of these questions may be addressed by the paper somewhere, but the fact that I still have these questions after multiple readings shows that the paper's organization really needs improvement (not just minor local changes) and the responsibility isn't entirely on the reader's part.

There are multiple typos/grammar mistakes that the authors should fix.
"to decomposes the", "in complexer scenarios"

[1] Yang et al. "Hierarchical Cooperative Multi-Agent Reinforcement Learning with Skill Discovery." AAMAS. 2020.

**Summary Of The Paper:**

This paper designs a new agent with the ability to estimate the local observations and potential goals of other agents, conduct message passing of such local estimates with neighboring agents, then use the outcome of message passing in a two-level hierarchical policy to select its own goal and take primitive actions conditioned on that goal. Experiments and ablations were done in the cooperative navigation benchmark and a multi-sensor target coverage task, in comparison to previous multi-agent communciation baselines.

**Summary Of The Review:**

Technical and experimental work is impressive but the organization of writing and lack of clarity in certain key parts makes this paper fall under the necessary standards.

During rebuttal: authors have provided detailed answers to my questions and revised the manuscript to improve clarity.

---

> ### Author Response · Authors · 2021-11-23
> **Response to Reviewer V4LB (1)**
>
> We are thankful for your time and insightful comments. We are particularly grateful for your recommendations to improve the manuscript. We have revised the manuscripts accordingly for a better reading experience. We answer your questions as follows.
>
> - **Q1: What does "consensus" refer to? If the authors intend to say that the agent's prediction of the agent's goal matches the agent's actual chosen goal, then the authors need to verify this in their experiments.**
> A1: Thanks for your clarification. We did intend to say that agent $i$'s prediction of agent $j$'s goal $G_{i,j}^*$ matches agent $j$'s actual chosen goal $G_j$. We have supplemented the experiment results in Appendix. D. The accuracy of goal inference is 80.2% $\pm$ 1.4%, while the accuracy of random prediction is 50.1% $\pm$ 0.3%.
>
> - **Q2: Does an agent chooses many different sub-goals at various steps in an episode? How many primitive actions are taken per sub-goal? Is each sub-goal drawn from a predefined set of possible goals? Or is a sub-goal a latent variable whose meaning emerges via some kind of unsupervised learning? Or are there hand-designed rewards for each sub-goal?**
> A2: First, ToM2C focuses on the **Target-oriented Multi-Agent Cooperation problem**, where "the agents need to cooperatively reach the goals and keep specific relations among the agents and targets." So the sub-goals are the selected targets for the low-level executor. In CN, the sub-goals are the landmark to reach. In MSMTC, the sub-goals are the targets to track.
> In an episode, an agent chooses its sub-goals at each high-level step. Then it will take primitive actions for $K$ low-level steps to reach the sub-goals. In our experiments, $K=10$. There are no hand-designed rewards for each sub-goal. In fact, there is only a global team reward shared by all the agents. In CN, the team reward is "the sum of the negative distance of each landmark to its nearest agent." In MSMTC, the team reward is the "coverage rate of targets".
>
> - **Q3: The authors claim that CTDE methods are flawed because "cooperation collapse in spite of a slight change in the team formation,..." Which work has shown this? What is meant by "change in team formation"?**
> A3: We apologize for this inappropriate statement in the related work. What we intended to claim is that the agent without communication is hard to quickly adapt to unseen cooperators/environments. We have revised this part accordingly.
>
> - **Q4: "pose" $(\phi_1,...\phi_n)$ is undefined. It also seems way too specific to particular applications.**
> A4: In general, the pose indicates the location and rotation of the agent.
> Specifically, “Pose” refers to the location $(x,y)$ of each agent in CN, while it includes the location and direction of cameras in MSMTC, noted as $(x, y, yaw)$. For most target-oriented multi-agent cooperation problems, we can easily define the "pose" of agents. Moreover, it can be easily measured/estimated by different sensors (e.g., GPS, IMU, camera, lidar) in real-world scenarios. So it is not specific to particular applications. We have revised Section 3 (Page 4) in the manuscript to clarity this term.
>
> - **Q5: How are edge features defined ?**
> A5: In section 3.3, page 6, "In the end, we obtain the final edge feature $(E_{i,j}, h_{i,j})$, and compute the probabilistic distribution over the type of the edge." Here $(E_{i,j}, h_{i,j})$ is the edge feature we used.
>
> - **Q6: n-1 ToM nets or one single ToM net.**
> A6: Conceptually, the ToM Net for an agent is composed of $n-1$ sub-modules to model the other n-1 agents, shown as Fig. 2. In practice, we can let these n-1 sub-modules share the same network parameters since all the agents are homogeneous. We have clarified this point accordingly in the manuscript.
>
> - **Q7: Authors should define that auxiliary task immediately after introducing it, not 2 pages later. Also, the accuracy of this prediction needs to be shown in an experiment.**
> A7: Thanks for your valuable suggestion. We have revised the manuscript accordingly. The experiment results are supplemented in Appendix. D. The accuracy of the auxiliary task for the observation estimation task is 98.1% $\pm$ 0.3%, while the accuracy of random prediction is 50.4% $\pm$ 0.6%.
>
> - **Q8: What is an "effect", as in "node effect" and "edge effect"? This is not a standard term in the graph neural network literature.**
> A8: The term "effect" refers to the effect of the interactions between agents. So both node effect and edge effect are computed based on the inferred states of others. As we stated in section 3.3, $h_j= \Psi^{\rm node}(V_j, h_j, \sum_{k}h_{k,j}),  h_{j,k} = \Psi^{\rm edge}(h_j, h_k, h_{j,k})$. $V_j$ is the node feature, $h_j$ is the node effect, $h_{j,k}$ is the edge effect. Actually, we adopted the term "effect" because it was originally used in Interaction Networks (Battaglia et al, NIPS 2016, Citation:878).

---

> ### Author Response · Authors · 2021-11-23
> **Response to Reviewer V4LB (2)**
>
> - **Q9: What exactly is Communication Reduction (CR)? What are the inputs and outputs? Is it a binary classifier that decides whether to cut or not cut an edge?**
> A9: Communication Reduction(CR) is a training process to guide the agents to learn to remove the unnecessary connection in the communication networks. As we stated in the revised manuscript, we observe that the decision made by receivers do not always influence by the received messages, indicating that some connections are actually redundant in the GNN. Therefore, it is necessary for us to figure out the really valuable connections from the densely connected networks. The **message-sender network** outputs the probabilistic distribution over the discrete type of the edge ($P_{cut}+P_{retain}=1$), so the loss is the binary cross-entropy loss: $l*\log(P_{retain})+(1-l)*\log(P_{cut})$. $l$ is a binary pseudo label, generated by discretizing the effect of the received message. We have revised section 3.3 to clarify the details of Communication Reduction.
>
> - **Q10: What do authors mean by an "actor-critic feature"? The observation input to the policy network?**
> A10: Shown as Fig.2, the actor-critic feature $\eta_i$ is an aggregation of inferred goals $G_i^*$, encoded observation $E_i$ and received messages $\sum g^*_{?,j}$, used for high-level planning (the choice of sub-goal). The concatenation of actor-critic features $(\eta_1,...\eta_n)$ among all agents is the input to the critic. We have revised the paper to avoid confusing terms.
>
> - **Q11: What is the reward for the goal-selector? What is the reward for the low-level executor? By goal and "sub-goal", do you mean the same thing? What is the reward corresponding to each sub-goal?**
> A11: Here goals and sub-goals are the same things, referring to the targets chosen by the high-level planner. The goal-selector is the high-level policy, which is trained according to the team reward. The low-level executor is trained to reach the goal with a goal-conditioned reward. In CN, the low-level reward is the negative distance to the selected landmark. Following HiT-MAC, the low-level reward in MSMTC is about the tracking errors to the selected targets.

---

> ### Comment · Reviewer_V4LB · 2021-11-27
> **Author's response has addressed my questions**
>
> I thank the authors for providing detailed answers to my questions and revising the manuscript to improve clarity. I will increase the score accordingly. There is a lingering sense that the exposition still has room for improvement, given the abundance of notation and interaction between many components in the whole algorithm. But I acknowledge that may be difficult due to the page constraints. Having a table of symbols and their meaning in the appendix may help future readers. Interestingly, the terms "node effect" or "edge effect" do not appear in Battaglia's survey paper "Relational inductive biases, deep learning, and graph networks".

---

### Official Review · Reviewer_jxPG · 2021-11-02

**Correctness:** 3
**Technical Novelty And Significance:** 2
**Empirical Novelty And Significance:** 3
**Recommendation:** 6
**Confidence:** 3

**Main Review:**

The paper was clearly written and easy to understand. The authors presented their motivations well. While using ToM inspired models has been explored in 2-agent tasks, integrating these models in a multi-agent setting to modulate communication was quite neat. Correspondingly, I found the ablation studies showing the utilities of ToM and the communication reduction to be very useful. I was particularly impressed by the model's generalization across different task settings and scales (different numbers of cameras and targets).

One of the weaker points of the paper was the use of privileged information to learn the ToM module; the ToMnet uses supervision from ground truth observations from other agents and requires supervision from their goal states. Access to others' true observations and goal states is unrealistic. Further, this reduces the extent to which the model is decentralized during training (as observation and goal states are directly used for supervision). Finally, the ToM model is limited in its novelty, but its use to influence communication is novel.

The authors evaluate the approach on two synthetic tasks: cooperative navigation and multi-sensor target coverage. I would have liked the method to be evaluated in more complex settings (as the authors themselves note in the conclusion) like SMAC [1] or the Hanabi Challenge [2](that specifically test for decentralized cooperation). If the authors believe that their method is inapplicable to the domains, then I would have liked to see a discussion on how this approach could be adapted.

One of the other lacking aspects of the paper was a discussion of other hierarchical baselines for modelling agents. I would have liked to see a discussion on such approaches. For example, [5] use a similar hierarchical approach for cooperation in a two-agent setting.

Finally, what do the authors think about approaches in IRL/ meta-IRL;  that directly try to predict the goals/ rewards of an agent?

I have added a few relevant references that work with agency reasoning and cooperation; I believe they are related to theory of mind inferences and collaboration. The authors might include them if they think that they are relevant.

Agency reasoning benchmarks: AGENT [3] BIB [4]

Cooperation benchmarks: Watch2Help [5] Overcooked [6] PHASE [7]

[1] Samvelyan, M., Rashid, T., De Witt, C. S., Farquhar, G., Nardelli, N., Rudner, T. G., ... & Whiteson, S. (2019). The starcraft multi-agent challenge. *arXiv preprint arXiv:1902.04043*.

[2] Bard, N., Foerster, J. N., Chandar, S., Burch, N., Lanctot, M., Song, H. F., ... & Bowling, M. (2020). The Hanabi challenge: A new frontier for ai research. *Artificial Intelligence*, *280*, 103216.

[3] Shu, T., Bhandwaldar, A., Gan, C., Smith, K. A., Liu, S., Gutfreund, D., ... & Ullman, T. D. (2021). AGENT: A Benchmark for Core Psychological Reasoning. *arXiv preprint arXiv:2102.12321*.

[4] Gandhi, K., Stojnic, G., Lake, B. M., & Dillon, M. R. (2021). Baby Intuitions Benchmark (BIB): Discerning the goals, preferences, and actions of others. *arXiv preprint arXiv:2102.11938*.

[5] Puig, X., Shu, T., Li, S., Wang, Z., Liao, Y. H., Tenenbaum, J. B., ... & Torralba, A. (2020). Watch-and-help: A challenge for social perception and human-AI collaboration. *arXiv preprint arXiv:2010.09890*.

[6] Carroll, M., Shah, R., Ho, M. K., Griffiths, T., Seshia, S., Abbeel, P., & Dragan, A. (2019). On the utility of learning about humans for human-ai coordination. *Advances in Neural Information Processing Systems*, *32*, 5174-5185.

[7] Netanyahu, A., Shu, T., Katz, B., Barbu, A., & Tenenbaum, J. B. (2021). PHASE: PHysically-grounded Abstract Social Events for Machine Social Perception. *arXiv preprint arXiv:2103.01933*.

**Summary Of The Paper:**

The paper presents a new method for communication and cooperation in multi-agent settings. The method relies on modelling other agents' intentions and internal states using Theory of Mind based neural nets. The predictions from the ToM model are used to decide how to communicate and coordinate with other agents. The authors test the method on two common multi-agent cooperation tasks to achieve SOTA communication efficiency and reward performance. The authors also show the utility of modelling mental states and using communication through ablation studies. Finally, the model shows flexibility in generalization, with consistent performance across different settings.

**Summary Of The Review:**

The use of a TOM model to facilitate communication for cooperation was impressive. I believe the paper is slightly below the acceptance threshold in its current state due to a lack of tests on more complex benchmarks that test for decentralized cooperation (SMAC/ Hanabi) and limited applicability because of access to unrealistic privileged information during training.

---

> ### Author Response · Authors · 2021-11-23
> **Response to Reviewer jxPG**
>
> We are thankful for your time and insightful comments. We are particularly grateful for your recommendations to improve the manuscript. We answer your questions and concerns as follows.
>
> - **Q1: The ToMnet uses supervision from ground truth observations from other agents and requires supervision from their goal states. Access to others' true observations and goal states is unrealistic. Further, this reduces the extent to which the model is decentralized during training (as observation and goal states are directly used for supervision). Finally, the ToM model is limited in its novelty, but its use to influence communication is novel.**
>
> A1: First, Thank you for your acknowledgment of the novelty of using ToM to influence communication. Note that additional supervision is only provided during the training process. In other words, ToM2C adopts Centralized Training Decentralized Execution framework(CTDE), which is widely leveraged in most popular MARL methods. So it is reasonable to employ the true observation and goal states for training.
>
> It is a good point to discuss the potential of extending our methods in fully decentralized training setting. Intuitively, we can consider enabling the agents to provide additional feedback (e.g., the inferred observation/goal is correct or not) via communication channel to other agents while training.
>
>
> - **Q2: The method to be evaluated in more complex settings, like SMAC [1] or the Hanabi Challenge [2].**
>
> A2: As we claimed in the introduction, ToM2C focuses on **Target-oriented Multi-Agent Cooperation (ToMAC) problem**, where "agents need to cooperatively adjust the relations among the agents and targets to reach the expectation". Hanabi is out of the scope, as there is no explicit target in the environment. It is feasible to deploy ToM2C on SMAC, where the opponents are regarded as the targets. In this case, the agents should select which one to attack at the high-level policy, and learn a low-level policy to reach and attack the target.
>
> Focusing on ToMAC, we argue that SMAC does not introduce any additional challenges in the target selection or the ToM inference, compared with the two environments we used. In CN, targets (landmarks) are static and the agent only needs to choose one target as sub-goal. In MSMTC, a more difficult setting, all targets are moving and each agent chooses multiple targets concurrently as its sub-goals. Besides, the agents are of limited observation range and the targets are frequently occluded by obstacles. Therefore, we argue that CN and MSMTC cover the main challenges in ToMAC: static targets vs. dynamic targets, single-choice vs. multi-choice sub-goal. As for SMAC, it can be regarded as a case that combines dynamic targets and single-choice sub-goal, which is not beyond the scope. So we do not think that the evaluation on SMAC should be placed in the first place. Of course, we agree that testing our method in SMAC can increase the impact of this work, since SMAC is a widely used benchmark for MARL algorithms evaluation. But limited by the time, we are sorry for the missing of additional results on SMAC in this time. In the revised Section 5, we add an additional discussion about evaluating ToM2C in other environments.
>
>
> - **Q3: Discussion of other hierarchical baselines for modelling agents.**
> A3: Most of the hierarchical methods share a similar spirit, aiming at decomposing complex task structure into multi-level tasks. Our hierarchy is also motivated by this. In this paper, we focus on designing a ToM-inspired communication mechanism among high-level policies to further enhance the multi-agent cooperation. We have revised the related work in multi-agent communication and cooperation to discuss it.
>
> - **Q4: What do the authors think about approaches in IRL/ meta-IRL; that directly try to predict the goals/ rewards of an agent?**
> A4: Good question! It will be feasible and interesting to employ IRL to learn the policy. As far as we know, most IRL methods require some expert demonstrations for learning. So the main challenge for us is how to collect a number of good demonstrations about multi-agent cooperation. Intuitively, the heuristic search policy with the global state can be a choice. However, we observed that it is costly in time to execute global searching in large-scale scenarios, e.g., it takes 4 minutes per step to find a near-optimal joint actions in 10 vs. 10 MSMCT. But it will be an interesting future direction to this work.
>
> - **Q5: References**
> A5: We have included all the mentioned papers in the revision. Thanks for your kind suggestion.

---

> > ### Comment · Reviewer_jxPG · 2021-11-27
> > **Re: Author response and revisions**
> >
> > I thank the authors for their detailed response. While testing on additional environments would have provided more validity to the framework, the central ideas introduced in the paper are interesting by themselves. With the revision addressing most of my concerns and other reviewers' concerns, I have raised my score.

---

### Official Review · Reviewer_YARZ · 2021-11-03

**Correctness:** 3
**Technical Novelty And Significance:** 3
**Empirical Novelty And Significance:** 2
**Recommendation:** 6
**Confidence:** 4

**Main Review:**

Strength
Overall model design to address the problem is interesting. In addition, performing two tasks (CN and MSMTC) for performance evaluation and choice of the baseline is sound. The ablation studies show how well the proposed model is structured. Code is provided to improve reproducibility.

Concerns
1.	As written in section 3, the proposed TOM2C receives a local partial observation. However, obtaining the current pose (guess that it is a ‘position’) of all agents does not seem to be a partial observation setting. More clarification should be provided about the partial observation.
 2.	The authors claim that TOM2C can provide fully distributed execution. What about the learning process? Does TOM2C consist of centralized learning?
3.	The paper does not state “who” provides rewards to all agents. In section 4.1, the authors ‘punish’ the team with a penalty, which implies the existence of a coordinator. Does TOM2C have a coordinator who gives rewards to all agents? Does it mean a centralized training framework?
4.	In section 3.1, the global information is an input to the attention module and partial observation is an output. Is the attention model a part of the agent? If then, can we still say the partial observation? The input to the agent will be the global information. If the attention is a part of the environment, does the environment have an individual attention module for each agent? How can it be explained in a real-world scenario?
5.	In section 4, the simulation results include the case of 3 agents. What happens if the number of agents increases? Does the proposed solution work in any multiagent system?
6.	Section 4.1 describes the cooperative navigation task and multi-sensor target coverage task. However, the details of tasks are missing. How much do the targets move in a single step? It would be much convincing if results of other benchmarks are provided. For example, Hanabi could be a great candidate to evaluate the performances of the MARL algorithm.
7.	In research on Theory of Mind, inference performance of ToM net is a significant factor. So, it would be beneficial to show the accuracy of the ToM net as an individual model, not as a part of ToM2C. In addition, it would be interesting to show the performance change of the entire ToM net according to the number of agents.


**Summary Of The Paper:**

This paper proposes a new algorithm called TOM2C to solve the multi-agent reinforcement learning problem. To achieve goals in the MARL problem, communication between agents is important. However, it is often challenging due to scalability and communication costs. To solve this problem, the authors adopt the Theory of Mind to multi-agents. The agent infers the mental states and intentions of others upon partial observation. TOM2C has two kinds of agents: a planner that decides sub-goal and reaches a consensus, and a low-level executor that takes actions. The authors also provide a communication reduction method based on CTDE.

**Summary Of The Review:**

The approach that combines MARL and Theory of Mind is interesting. The proposed TOM2C can be applied to multiple MARL problems which need cooperation between agents. But the proposed algorithms should be explained more in detail. Also, more experiments are needed to persuade the reviewers regarding its performance.

---

> ### Author Response · Authors · 2021-11-23
> **Response to Reviewer YARZ**
>
> Thank you for your detailed comments and helpful suggestions, we are encouraged that you value the novelty of the work. We clarify some misunderstandings and answer your specific questions in the following comments. Besides, we have also revised our manuscript accordingly.
>
> - **Q1: The details about pose and partial observation.**
> A1: In general, the pose indicates the location and rotation of the agent. It can be easily measured/estimated by different sensors (e.g., GPS, IMU, camera, lidar) in real-world scenarios. Specifically, "Pose" refers to the location $(x,y)$ of each agent in CN, while it includes the location and direction of cameras in MSMTC, noted as $(x,y,yaw)$. Accessing the poses of other cooperators does not violate the partial observation setting. That is because the state of the target is still partially observable to the agents, i.e., agents still can not observe the states of the targets that are out of view.
>
> - **Q2: Does TOM2C consist of centralized learning?**
> A2: Yes, ToM2C adopts Centralized Training Decentralized Execution framework (CTDE), which is widely leveraged in MARL. The training of ToM net and critic requires global information. We have revised Section 3.5 to clarify this point.
>
> - **Q3: Does TOM2C have a coordinator who gives rewards to all agents? Does it mean a centralized training framework?**
> A3: There is no centralized coordinator in our method. The team reward (or punishment) comes from the environment. For example in MSMTC, as we stated in Section 4, "the reward is the coverage rate of targets." All agents share this team reward while learning. We have revised Section 3.5 to clarify this point.
>
> - **Q4: Is the attention module a part of the agent or the environment?**
> A4: First, we apologize for the ambiguous term "global information". In fact, this global information input refers to all the targets in the observation field of an agent, instead of the global information of the environment. Each agent has an attention module for observation encoding. With this attention module, it can fuse the state of the observed targets (in its view) into a single feature. We have revised Section 3.1 to replace the confusing term.
>
>
> - **Q5: What happens if the number of agents increases?**
> A5: In Section 4.4, we conducted the scalability experiment and reported the results among 81 settings (from 2 vs. 2 to 10 vs. 10) in Figure 6. We can see that the performance of ToM2C is rather stable when the number of agents and targets changes, which means that ToM2C has a good generalization.
>
> - **Q6: Does the proposed solution work in any multiagent system?**
> A6: As we claimed in the introduction, ToM2C focuses on **Target-oriented Multi-Agent Cooperation problem**, where "agents need to cooperatively adjust the relations among the agents and targets to reach the expectation". Such problem setting widely exists in real-world applications, e.g., collecting objects, navigating to landmarks, transporting objects, and monitoring a group of moving objects.
>
> - **Q7: Details of the tasks are missing. How much do the targets move in a single step?**
> A7: Limited by the page constraints, we provided the details of MSMTC and CN in Appendix. A. In MSMTC, targets move in the direction according to their type (destination-navigation or random walking). At each step, the target samples a velocity from a pre-defined range and moves in the type-related direction. In CN, targets do not move at any time.
>
> - **Q8: It would be much convincing if results of other benchmarks are provided. For example, Hanabi could be a great candidate to evaluate the performances of the MARL algorithm.**
> A8: As we explained in A6, ToM2C is proposed to solve the Target-oriented Multi-Agent Cooperation problem. Hanabi is out of the scope. We evaluated our methods on two representative target-oriented problems. Cooperative Navigation (CN) represents the simplest setting, where targets are static and the agent only needs to choose one target as its sub-goal. The multi-sensor multi-target coverage (MSMTC) problem represents a more difficult setting, where targets can move randomly and occluded by the obstacles, but the locations of sensors are fixed during an episode. In this task, all targets are dynamic and each agent chooses multiple targets concurrently as its sub-goals.
>
> - **Q9: Show the accuracy of the ToM net as an individual model, not as a part of ToM2C. Show the performance change of the entire ToM net according to the number of agents.**
> A9: We have supplemented the results of ToM accuracy in Appendix. D. The accuracy of goal inference is 80.2% $\pm$ 1.4%. The accuracy of observation estimation is 98.1% $pm$ 0.3%. For comparison, the accuracy of random prediction are 50.1% $\pm$ 0.3% and 50.4% $\pm$ 0.6% respectively. Note that the accuracy of ToM prediction is stable when the number of agents changes.

---

> > ### Comment · Reviewer_YARZ · 2021-11-29
> > **authors' response**
> >
> > The authors' response was helpful in general. However, the reviewer still has a question regarding the validation of partial observation settings, i.e., the agent has partial observation toward the target but full observation toward other agents, which makes the task simpler. Since I gave the highest initial score, I will keep my current score.

---

> > > ### Author Response · Authors · 2021-11-29
> > > **Further validation**
> > >
> > > Thanks for your reply and suggestions! To address your concern about the environment setting, we conducted additional validation by adding a restriction on the access of agent poses, i.e, agents can only access the poses of others within an observable distance. In this case, the agent is also partially observable to others, i.e., if agent $j$ is out of the view of agent $i$, we mask the ToM inference and communication to agent $j$ from agent $i$. Under such partially observable agent poses, we tested our model in MSMTC. We reported the average reward among 100 episodes in the following Table, comparing with the case of using all agent poses.
> > >
> > > | sensors vs. targets      |      Partially observable agent poses       |    All agent poses  |
> > > | :---: | :------------: | :-----------: |
> > > | 10 vs. 10 | 75.96 $\pm$ 0.81  | 77.26 $\pm$ 0.52 |
> > > | 10 vs. 5  | 85.06 $\pm$ 0.32  | 84.69 $\pm$ 0.77 |
> > > |  4 vs. 5  | 74.73 $\pm$ 0.95 | 75.38 $\pm$ 0.57  |
> > >
> > > From this Table, we can see that the average reward of ToM2C is still comparable to the original version (Use all agent poses) when the poses of agents are partially observable. Note that in 10 sensors settings, only ~4 agents are observable to each agent at each step on average. These validated that ToM2C is of good generalization in case that other agents are partially observable.

---

### Author Response · Authors · 2021-11-24
**Change logs**

We updated our paper during the rebuttal period, which could be summarized as below:
- (1)  we revised the Introduction to clarify some concepts and points for a better reading experience, such as the definition of ToMAC and illustration of experiment tasks.
- (2) We revised the Related Work section to enrich the discussion on hierarchical methods and agency reasoning benchmarks. Moreover, we replaced the inappropriate statement about CTDE methods.
- (3) We fixed the notations in Figure 2 and Section 3 to keep them consistent.
- (4) We adjusted the Organization of the Methods section for a better reading experience. The part of learning ToM Net was moved from Training section to the end of ToM Net section. Besides, we edited the introduction of message-sender to make it clearer.
- (4) We clarified the Details of ToM2C model, such as the number of ToM nets in a single agent model, the definition of auxiliary task, pose and edge feature, the input of attention module.
- (5) We revised the explanation of Communication Reduction and added the corresponding loss function formula.
- (6) We merged the original Section 4.1 (Task and Setups) with the first paragraph of Section 4 (Experiments), considering the page limitation. More details on environments were moved to Appendix A. We also added the explanation of the type of ToMAC (static/dynamic targets,single/multi choice sub-goal) and the choice of environments.
- (7) We added more discussions on extending ToM2C to other scenarios (such as SMAC and Hanabi) in Section 5.
- (8) We added additional experiment results of ToM accuracy in the end of Appendix. B.

---

### Decision · Program_Chairs · 2022-01-20

**Decision:**

Accept (Poster)

**Comment:**

The current paper presents a new method for communication and cooperation in multi-agent settings. Specifically, the authors propose to model other agents' intentions and internal states using ToM nets and using these predictions to then decide how to communicate/coordinate. The authors present experiments in two multi-agent cooperation tasks (multi-sensor multi-target coverage and cooperative navigation), compare against 4 previous methods (TarMAC, I2C, MAPPO and HiT-MAC) and perform the necessary ablations studies and find that their method achieve better rewards in both environments.
All reviewers have found the present study to be novel with convincing experimental findings. Reviewers have raised some concerns however a great deal of those have been addressed by the authors during the rebuttal and many of these points have now been incorporated in the paper.

Having read the paper and considering the reviews I agree with the reviewers that this manuscript will make a good addition to the program of ICLR and as such I recommend its acceptance.